# Manhole Cover Classification Based on Super-Resolution Reconstruction of Unmanned Aerial Vehicle Aerial Imagery

Dejiang Wang * and Yuping Huang

School of Mechanics and Engineering Science, Shanghai University, Shanghai 200444, China; 1873835005@shu.edu.cn
* Correspondence: djwang@shu.edu.cn

**Abstract:** Urban underground pipeline networks are a key component of urban infrastructure, and a large number of older urban areas lack information about their underground pipelines. In addition, survey methods for underground pipelines are often time-consuming and labor-intensive. While the manhole cover serves as the hub connecting the underground pipe network with the ground, the generation of underground pipe network can be realized by obtaining the location and category information of the manhole cover. Therefore, this paper proposed a manhole cover detection method based on UAV aerial photography to obtain ground images, using image super-resolution reconstruction and image positioning and classification. Firstly, the urban image was obtained by UAV aerial photography, and then the YOLOv8 object detection technology was used to accurately locate the manhole cover. Next, the SRGAN network was used to perform super-resolution processing on the manhole cover text to improve the clarity of the recognition image. Finally, the clear manhole cover text image was input into the VGG16_BN network to realize the manhole cover classification. The experimental results showed that the manhole cover classification accuracy of this paper's method reached 97.62%, which verified its effectiveness in manhole cover detection. The method significantly reduces the time and labor cost and provides a new method for manhole cover information acquisition.

**Keywords:** image super-resolution reconstruction; manhole cover recognition; manhole cover positioning; drone aerial images

## 1. Introduction

Underground pipe networks play a crucial role in the daily operation of cities. As part of the urban municipal system, they cannot be ignored in the process of urban intelligent digitization. With the continuous development of smart cities and the continuous enrichment of urban functions, the process of municipal public facilities construction is gradually accelerating. This means that municipal, power, communications, and other departments need more practical measures to manage a large number of municipal equipment and assets.

At present, many old urban areas have a lack of information on underground pipelines, which makes it impossible to carry out intelligent management of pipelines. In turn, the precise lack of access to underground pipelines is a challenge for urban infrastructure management. Although a variety of techniques, such as ground penetrating radar (GPR), acoustic detection, stratigraphic detection, geomagnetic detection, and holographic interferometry, exist for the detection of underground pipe networks [1–3]. These methods usually require a large investment of labor and resources, which limits their efficiency and usefulness. These traditional methods are labor intensive and often lead to excessive consumption of time and cost. Manhole covers, serving as critical connectors between the underground pipeline network and the surface, play an essential role; thus, mapping the

layout of the underground network can be achieved by acquiring information on the type and location of manhole covers.

As artificial intelligence technology progresses and the use of unmanned aerial vehicles (UAVs) becomes more widespread in engineering applications, more and more deep learning techniques are being leveraged to solve engineering-related problems [4–6]. For example, many researchers have applied object detection algorithms to the task of locating manhole covers. Ying [7] and colleagues had introduced a method termed MA-FPN, which integrated the core concepts of attention mechanisms with feature pyramid networks, aiming to improve the accuracy of object detection in remote sensing. Liu et al. [8] designed a diversity feature extractor based on a VGG feature extractor to improve the performance of manhole cover localization by increasing the size of the receptive field. These methods obtained the city image by using UAV aerial photography, and then used the object detection technology to detect the aerial image to obtain the localization of the manhole cover. It greatly reduced the consumption of manpower and material resources, and also improved the survey efficiency. However, these methods had not solved the problem of obtaining the category information of manhole covers, and could only obtain the location information of manhole covers. However, in the process of urban digitization, the category information of manhole cover is indispensable. Only by correctly obtaining the category information of manhole cover can we build a complete municipal pipe network system. Other scholars have proposed using vehicles equipped with cameras and lidars to collect ground images and combine target detection methods to obtain manhole cover information [9,10]. For example, Wei et al. [11] used a combination of multiple symmetrically arranged cameras and lidars to classify manhole covers through descriptors and support vector machine algorithms. Pang et al. [12] proposed a real-time road manhole cover detection method based on deep learning model. By optimizing the network structure and reducing the size of the model and the number of parameters, the effect of deployment on vehicle-mounted embedded devices is achieved. Mattheuwsen et al. [13] developed a fully automatic method for manhole cover detection using mobile mapping point cloud data. Although the above method can solve the problem of manhole cover classification, due to the limitation of vehicle-mounted tools, a large number of manhole covers are located in places where cars cannot reach. It is also labor-intensive and time-consuming to collect ground images through radar- and camera-equipped vehicles, which are still inefficient methods. However, it is a feasible way to classify manhole covers after obtaining ground images by UAV. How to realize the classification of manhole covers is the main problem to be solved in this paper. Classification of manhole covers is mainly realized by recognizing the text on the cover; however, the manhole covers acquired from aerial images often lose the detail information of the text on the cover, so directly classifying manhole covers by target detection is not a feasible method. Image super-resolution reconstruction methods were designed to enhance the detail performance of low-resolution images. By recovering high-frequency detail information from low-resolution images, the image becomes clearer and has a higher detail resolution. These methods are widely used in the fields of aeronautics, medicine, and engineering [14,15]. In the field of aviation, a variety of image super-resolution reconstruction algorithms had been proposed [16–20]. Zhou et al. [18] proposed a super-resolution reconstruction strategy based on self-attention generative adversarial networks, which improves the details of remote sensing images by adding self-attention modules. Yue et al. [21] proposed an improved enhanced super-resolution generative adversarial network (IESRGAN) based on enhanced U-Net structure, which is used to perform a four-fold scale detail reconstruction of LR images using NaSC-TG2 remote sensing images. The above method can improve the detail of aerial images well. Therefore, it becomes possible to recover the details of manhole cover text by reconstructing the manhole cover from aerial photographs through super-resolution reconstruction network.

Based on the above analysis, this paper proposed a method to localize and classify manhole covers based on super-resolution generative adversarial network reconstruction of UAV aerial images. Initially, the YOLOv8 object detection network was employed to localize

manhole covers. Post-localization, these covers were segmented from the captured images. Further text localization was conducted on the segmented images using YOLOv8. This was followed by the application of the SRGAN super-resolution network to enhance the image quality of the manhole covers. The final step involved classifying the manhole covers using the VGG16_BN image classification network. The method used in this paper can greatly improve the efficiency of manhole cover survey and reduce the loss of manpower and material resources.

## 2. Methods

Among the many categories of manhole covers, the text on the cover is the most important feature to distinguish the cover. Classifying manhole covers by recognizing the text on the cover is the most effective way. This study introduces an approach for the classification of manhole covers. It began with the acquisition of aerial datasets through unmanned aerial vehicle (UAV) photogrammetry. Then, the object detection network, image super-resolution reconstruction network, and image classification network for the manhole cover classification were combined.

First, the ground aerial dataset is collected using UAV aerial photography, the collected data were cropped to the input size of the network, and the collected aerial images were labeled using labeling software to produce the dataset for manhole cover localization, and then the produced dataset was input into the object detection network for training. The trained network was used to localize the manhole cover and cut out the localized manhole cover. Next, the cropped manhole cover was used to create the text recognition dataset, then the object detection network was used to train the text localization network, and finally the text was localized and cropped according to the trained text localization network.

At the same time, high-resolution images of manhole covers were collected at low altitude using UAV, and after cropping and data enhancement, they were fed into an image super-resolution reconstruction network for training, and finally the cropped images of manhole covers were fed into the super-resolution reconstruction network to obtain a clear text of the manhole covers, and the clarified images of the text of the manhole covers were used to produce a dataset for classifying the manhole covers.

Finally, the manhole cover classification model was obtained by training the generated manhole cover classification dataset using VGG16_BN network. The specific technology roadmap is shown in Figure 1.

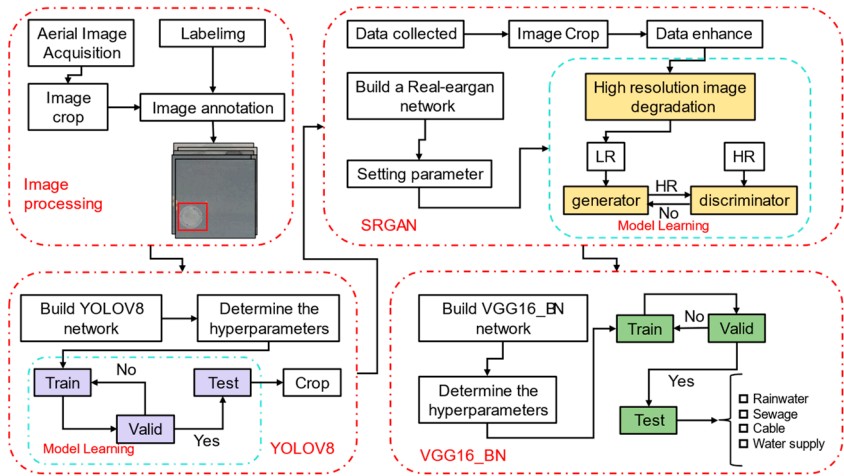

**Figure 1.** Flow chart for locating and classifying manhole covers.

### 2.1. Manhole Cover Positioning

In this paper, manhole cover classification is realized by recognizing the text on manhole covers. However, as text recognition in natural scenes, there are huge difficulties in directly utilizing aerial pictures for text recognition. Due to the complexity of the

image background obtained from aerial photography, a single image may contain multiple manhole covers and other text and other external factors interference [22]. Therefore, this paper compares the manhole cover localization effect of three target detection networks, YOLOv8, Faster R-CNN, and EfficientNet. The best network is selected and then the text is cropped out to avoid the interference of other manhole covers and other text in the scene.

### 2.1.1. YOLOv8 Network Architecture

YOLOv8 is composed of three parts: Backbone, FPN, and Yolo Head [23,24], as shown in Figure 2. The Backbone network uses CSPDarknet to extract the effective feature layer, and then realizes the feature fusion of the effective feature layer in FPN. By combining the feature information of different scales, the detection ability of FNP for targets of different sizes can be enhanced. Finally, the localization classification task is realized by Yolo Head. Because YOLOv8 network has strong detection ability for targets of different scales, YOLOv8 has great advantages in manhole cover detection task.

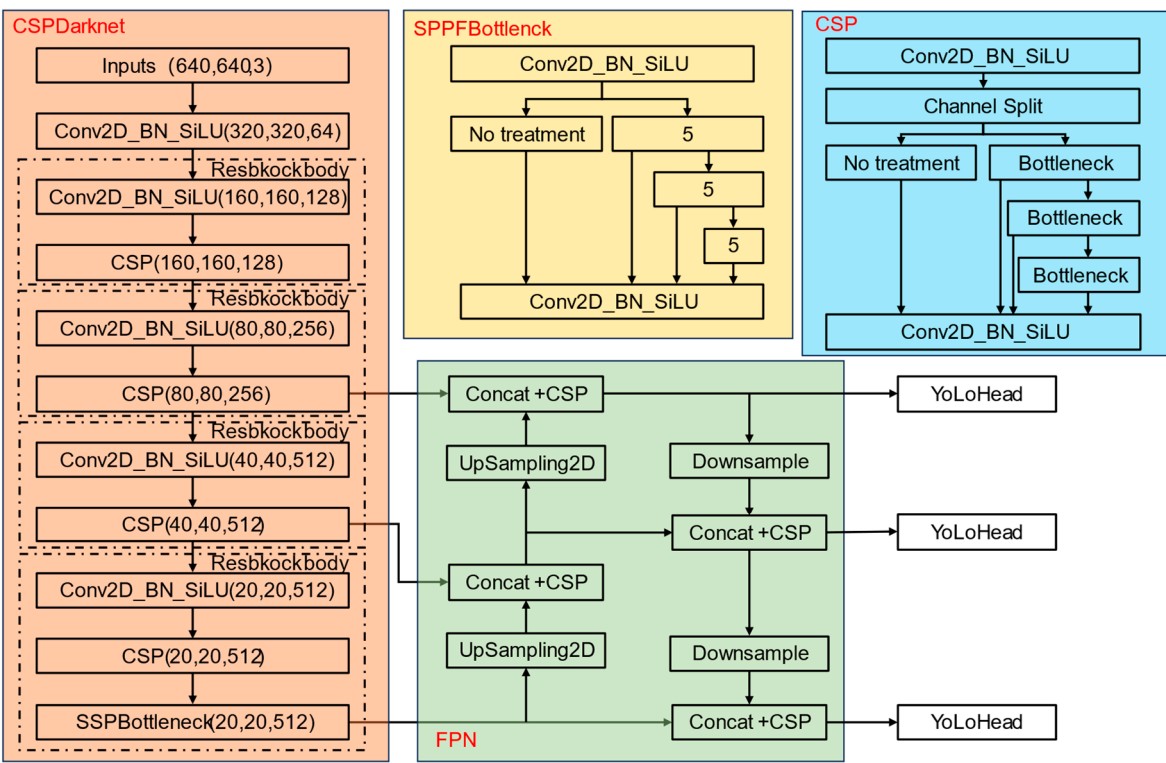

**Figure 2.** YOLOv8 structure diagram.

### 2.1.2. YOLOv8 Network Data Set Production

YOLOv8 network training was achieved by giving the detection box of the detection object and setting the corresponding labels for different objects. Therefore, it is necessary to make a manhole cover data set with annotation before network training. In this paper, aerial images were used to make a manhole cover positioning data set. There are many types of manhole covers, but for some special manhole covers, their categories do not need to be identified by text recognition, such as the rainwater grate in Figure 3, which can be distinguished by manhole cover positioning. Therefore, the manhole cover positioning only needs to pay attention to the categories of rainwater and sewage, rather than the image classification and recognition of the rainwater grate.

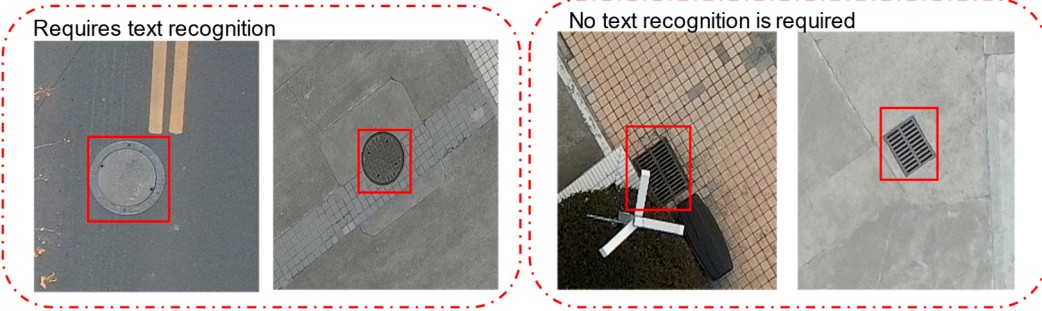

**Figure 3.** Manhole cover positioning data set display.

In the process of dataset production, it is necessary to cut the image according to the input size of 416 × 416 required by the network. As shown in Figure 4a. However, cutting the image directly into a 416 × 416 size will cause the manhole cover to be cut incompletely, eventually leading to the manhole cover category cannot be obtained. Because in the aerial image, the pixel value of the manhole cover is only approximately 80 pixels. Therefore, to ensure that the manhole cover is complete in at least one picture, it is only necessary to set the cutting overlap rate to 20%. The method of setting the overlap ratio is shown in Figure 4a. From Figure 4b, it can be seen that the left side of manhole cover 1 and 2 and 3 is complete by setting the overlap rate when cutting. After the cutting is complete, the aerial image is labeled by the target detection labeling software Labelimg 1.8.6, and the labeling result is shown in Figure 3. Finally, the dataset diversity is filled by means of data enhancement.

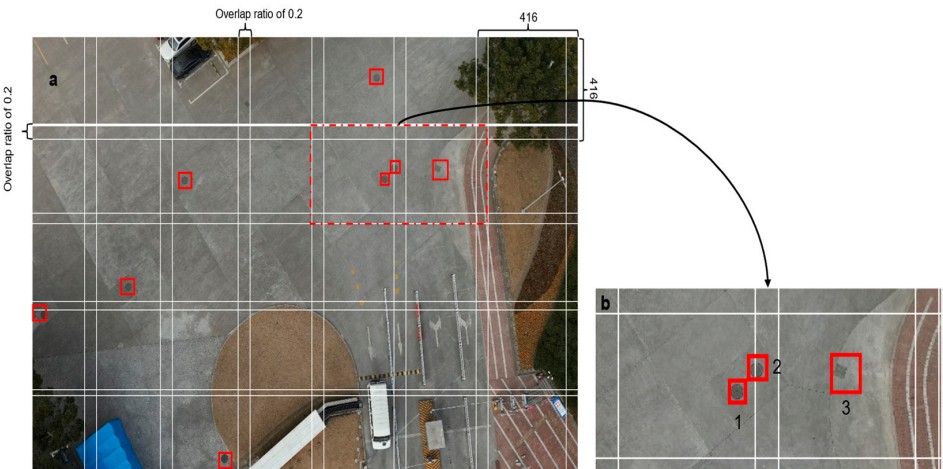

**Figure 4.** Picture overlap rate display. (**a**) cutout schematic, (**b**) avoiding manhole cover cutout schematic.

### 2.2. Text Localization

Character recognition first needs to locate and segment the text. Therefore, after obtaining the manhole cover image, the character recognition cannot be directly carried out [25]. It is also necessary to locate the text to obtain the text image. In this paper, the text positioning network still uses the YOLOv8 network.

Text Positioning Data Set Production

Text localization and manhole cover localization are the same need to mark the detection object location and label in advance. While the text localization dataset was the aerial image input into the trained manhole cover localization network, the manhole cover would be localized and cropped out. Cropping needed to be eliminated after the residual manhole cover does not contain the text, and then the LabelImg software was used to annotate the image.

*2.3. Super-Resolution Network Reconstruction Manhole Cover Text*

As text recognition in natural scenes, manhole cover text recognition is often affected by factors such as illumination changes and image sharpness, resulting in unclear text and the loss of many important features [26]. Due to the shooting distance and camera pixel limitations, manhole cover text captured by UAV loses a large number of details, as shown in Figure 5. The direct use of UAV images for manhole cover text recognition would result in a very low recognition accuracy, and the use of such fuzzy text for manhole cover classification training would result in the inability to obtain accurate manhole cover categories. Therefore, in this paper, SRGAN image super-resolution reconstruction technique was used to reconstruct the aerial images of manhole cover text at super-resolution, to recover the detailed features of manhole cover text, and to improve the accuracy of text recognition.

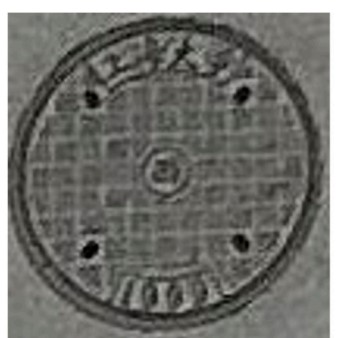
Blurred rain
manhole cover

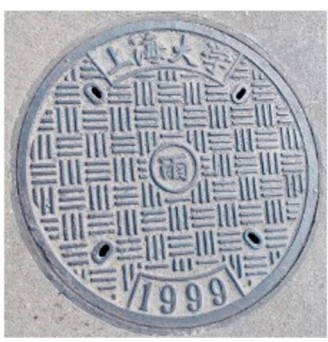
Clear rainwater
manhole cover

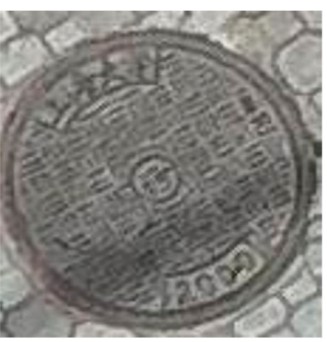
Blurred sewage
manhole cover

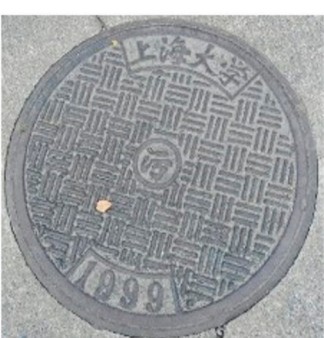
Clear sewage
Manhole cover

**Figure 5.** Fuzzy display of manhole cover.

2.3.1. SRGAN Network

Super-resolution generative adversarial network (SRGAN) is composed of generator and discriminator [27], as shown in Figure 6. The high-resolution degradation model is used to reduce the high-resolution manhole cover to a low-resolution image. The generator then upsamples the low-resolution image to generate a realistic high-resolution image and inputs it into the discriminator. The discriminator is used to distinguish the difference between the generated image and the real image to optimize the parameters of the image generated by the generator, and finally realize the super-resolution reconstruction of the manhole cover text.

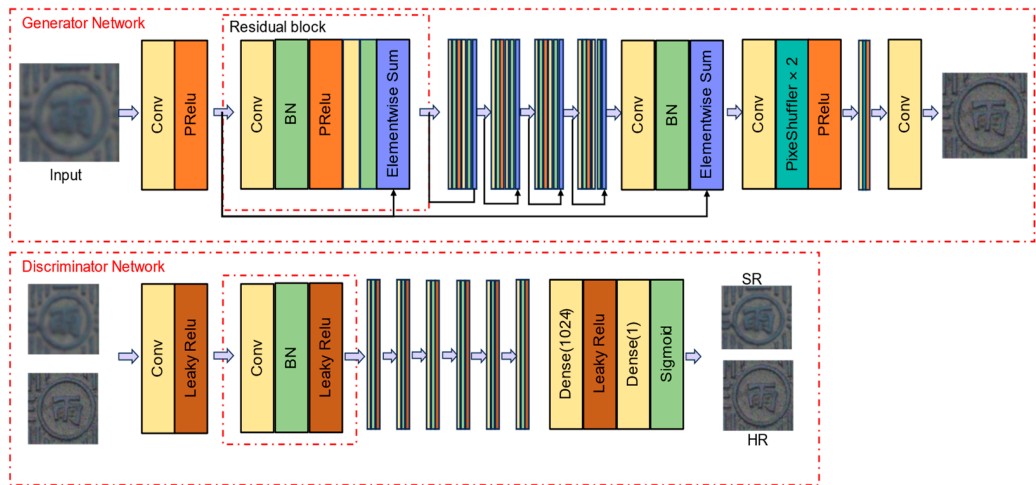

**Figure 6.** SRGAN network structure.

### 2.3.2. The Production and Enhancement of SRGAN Network Dataset

SRGAN, a single-image super-resolution reconstruction network, requires only the acquisition of high-resolution images. The corresponding low-resolution images are derived by downgrading these high-resolution images. Therefore, this paper described a process of capturing high-resolution images of manhole covers using unmanned aerial vehicles (UAVs) at low altitudes, followed by cropping the textual segments from these covers to compile the training dataset. To increase the variety within the dataset, data augmentation methods were employed, enhancing the dataset as illustrated in Figure 7. The specific data enhancement parameters are shown in Table 1:

Aerial pictures with different lighting conditions can be simulated by changing the brightness, while flipping can simulate the drone shooting from different directions. Moreover, noise and Gaussian blurring can simulate the drone's noise due to the shooting height and the environment and the image brought about by not focusing accurately.

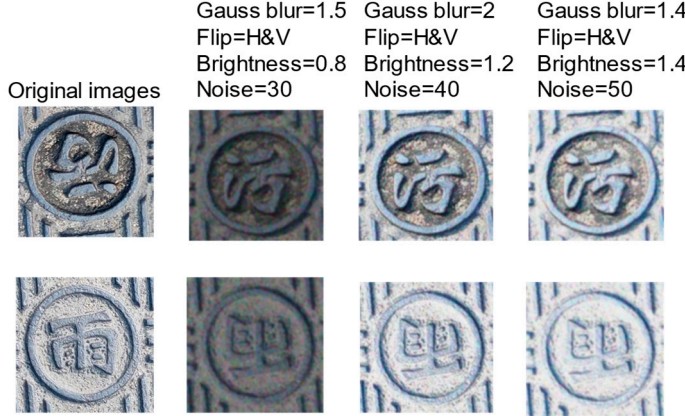

**Figure 7.** Enhanced display of text super-resolution reconstruction dataset.

**Table 1.** Data enhancement method and its corresponding parameters.

| Methods | Operation Execution |
| --- | --- |
| Brightness | Brightness factor = 0.8;1.2;1.4 |
| Flip | Horizontally flip; Vertically flip |
| Gauss blur | Blur radius =1.5;2;2.5 |
| Noise | Noise factor = 30;40;50 |

### 2.3.3. SRGAN Network Training

The training of the SRGAN network is conducted in two steps, as depicted in Figure 8. Initially, once the manhole cover text dataset had been compiled, the data were fed into the degradation model to produce low-resolution images. These images were then passed to the generator model for training, which synthesized high-resolution images. Subsequently, these images were submitted to the discriminator for assessment. Thereafter, the generator's parameters were refined based on the discriminator's evaluation of the authenticity of the images generated by the generator. Ultimately, this process results in the creation of high-resolution manhole cover text images.

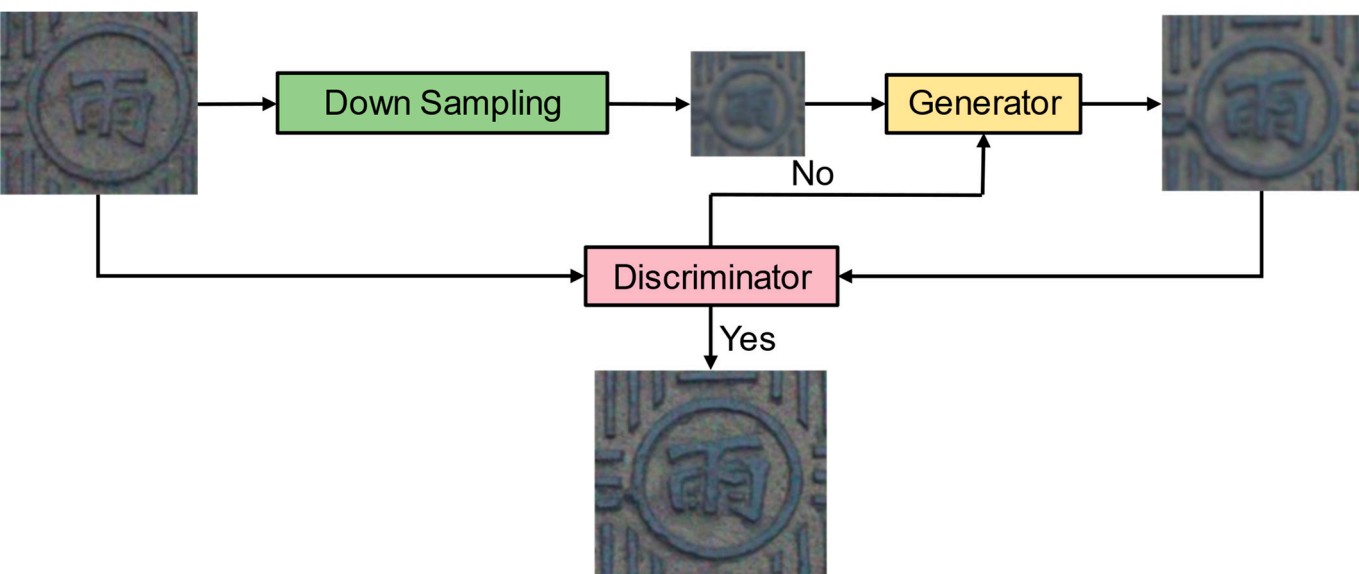

**Figure 8.** SRGAN network training process.

### 2.4. Image Classification Realizes Manhole Cover Classification

Traditional text recognition is achieved through optical character recognition (OCR) technology [28,29], whereas the text obtained from drone aerial photography loses a large number of textual features, and the text is not in the form of a standardized font. It is not feasible through traditional text recognition methods, and this can be ignored by using image classification networks for text recognition. Therefore, this paper compares three image classification networks, Mobilenetv1, Swin_transformer_tiny, and VGG16_BN, and selects the one with the best effect as the network for manhole cover classification.

#### 2.4.1. VGG16_BN Network Architecture

VGG16_BN is added to the batch normalization layer after each convolution layer of VGG16 [30]. As shown in Figure 9. The convergence speed of the network will be faster after the introduction of the batch normalization layer, and it has better generalization performance. Deeper feature information can be learned, which is very beneficial for manhole cover recognition. Therefore, the VGG16_BN network can more accurately identify the category of manhole cover.

#### 2.4.2. Text Recognition Data Set Production

Before training the VGG16_BN network, it is necessary to construct a dataset containing the text of the manhole cover. First, the text part of the manhole cover was extracted from the images captured by the UAV. Then, after determining the categories of manhole covers through field surveys, corresponding labels for manhole covers were set based on the surveyed categories of manhole covers. The dataset produced in this paper includes two categories of manhole covers, including rainwater manhole covers and wastewater manhole covers, and the categories are shown in Figure 10.

Image
(h×w×3)

Conv [1×1 ×16]
Conv [3×3 ×16]
Conv [3×3 ×32]
Conv [3×3 ×32]

Region score
(h/2×w/2 ×1)

Affinity score
(h/2×w/2 ×1)

VGG16-BN

Conv Stage1
(h/2×w/2×64)

Conv Stage2
(h/4×w/4×128)

Conv Stage3
(h/8×w/8×256)

Conv Stage4
(h/16×w/16×512)

Conv Stage5
(h/32×w/32×512)

Conv Stage6
(h/32×w/32×512)

UpConv Block
(h/2×w/2×32)
Stage4

UpSample(2x)
UpConv Block
(h/4×w/4×64)
Stage3

UpSample(2x)
UpConv Block
(h/8×w/8×128)
Stage2

UpSample(2x)
UpConv Block
(h/16×w/16×256)
Stage1

UpConv Block

Batch_norm
Conv [3×3 ×out_ch]
Batch_norm
Conv [3×3 ×(out_ch×2]

⊕ : Concat

**Figure 9.** VGG16_BN network architecture diagram.

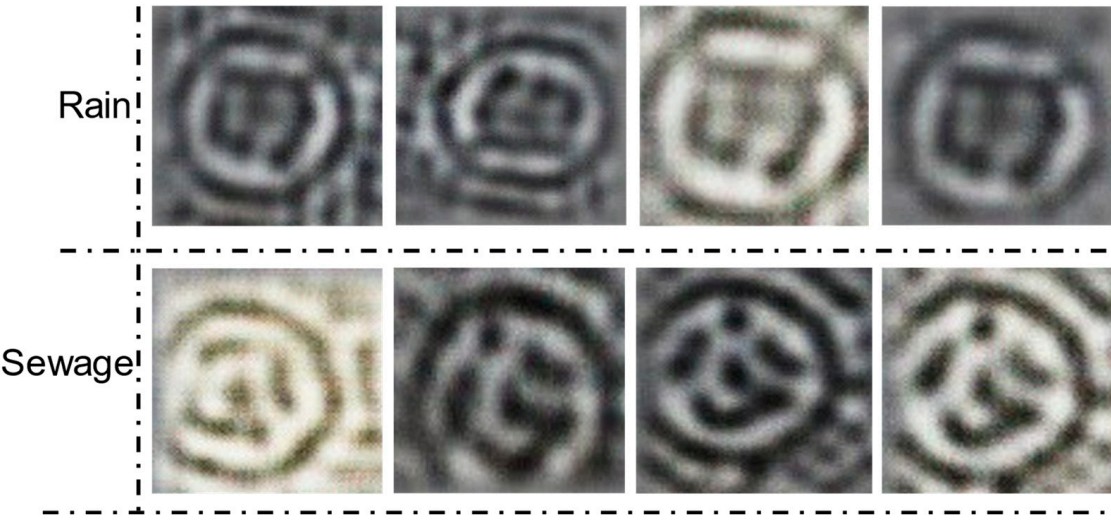

**Figure 10.** Category display of manhole cover classification.

In order to assess the consistency and reliability of the results of the classification, this paper uses the K-fold cross-validation method to divide the dataset for training, the K value is taken as 5, and the division schematic is shown in Figure 11.

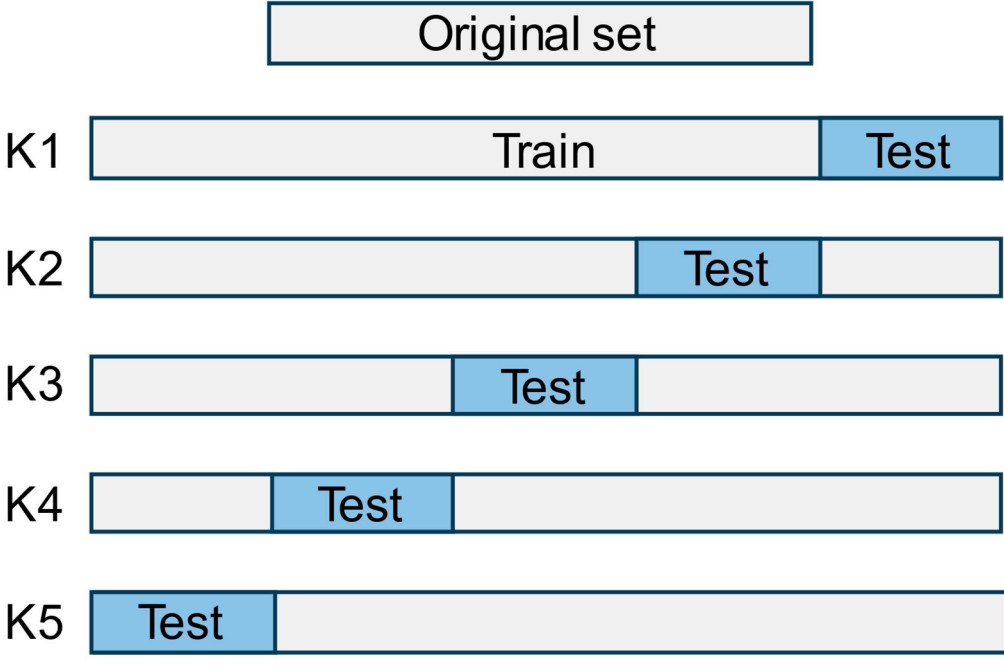

**Figure 11.** K-value cross-validation dataset division.

## 3. Experiments

### 3.1. Software and Hardware

This experiment mainly used aerial images to evaluate the scheme of the manhole cover classification network. The experiment is based on the deep learning framework proposed in this paper. It runs on a 13th Gen Intel (R) Core (TM) i7-13700KF CPU, the GPU is an NVIDIA 4090, it has 64 GB of memory, the Python version is 3.8, and the operating system is Windows 10.

### 3.2. Dataset Acquisition

3.2.1. Manhole Cover Positioning Data Set Acquisition

In this experiment, the DJI UAV M300 RTK equipped with the Zenmuse P1 camera (Manufacturer is DJI, company location is Shenzhen, China) was used for aerial photography at a university in Shanghai. The flight height was set to 50 m, with a heading overlap rate of 90% and a side phase overlap rate of 25%. To compare the experimental results under different light and weather conditions, the experiment was conducted on both sunny and cloudy days. Once the flight route was designed, the aerial photography was automatically performed. A total of 1276 photos were collected, with each picture having dimensions of 8192 × 5460 pixels, as shown in Figure 12. Among these, 999 pictures were taken on sunny days, and 277 pictures were taken on cloudy days. The resulting manhole cover positioning dataset consisted of 5122 images. Among these, 3420 pictures required classification, while 1702 pictures did not require positioning. The dataset was divided into training, testing, and validation sets in an 8:1:1 ratio.

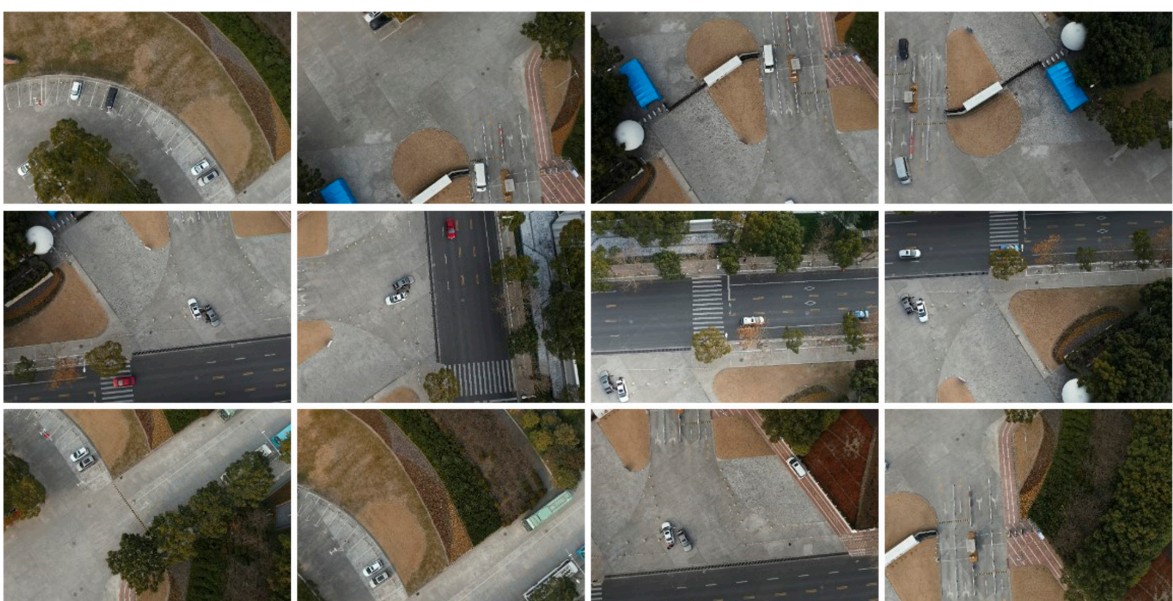

**Figure 12.** Aerial image of manhole cover positioning.

### 3.2.2. Text Location Data Set Acquisition

Manhole cover images are obtained by inputting aerial images into the trained manhole cover localization network, and a total of 3385 complete manhole cover images were collected. The dataset was divided according to the ratio of 8:1:1 for the training set, test set and validation set.

### 3.2.3. Manhole Cover Image Super-Resolution Data Set Acquisition

Text super-resolution reconstruction requires high-resolution manhole cover images, so it requires low-altitude UAV acquisition when collecting text super-resolution reconstruction data sets. In this paper, DJI UAV M300 RTK (Manufacturer is DJI, company location is Shenzhen, China) is used to carry Zenmuse P1 low-altitude shooting and acquisition. The flight height was set to 20 m. A total of 188 manhole covers were obtained, with a pixel size of $8192 \times 5460$, as shown in Figure 13. The pixel size after cutting is $71 \times 81$, including 109 rain manhole covers and 79 sewage manhole covers. The data set was expanded to 752 by means of data enhancement such as adding noise, Gaussian blur, tone change, and flipping.

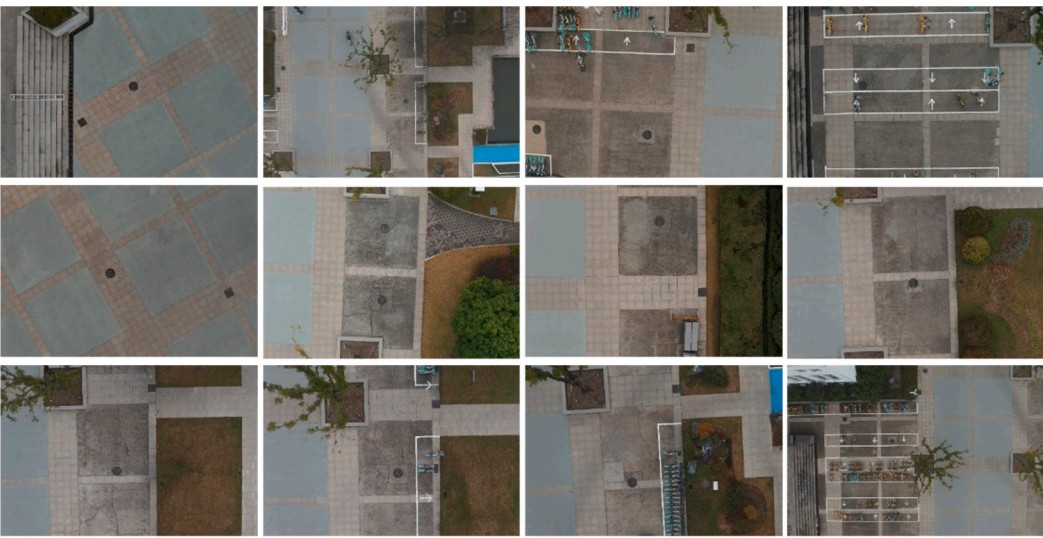

**Figure 13.** Super-resolution reconstruction of aerial image.

### 3.2.4. VGG16_BN Network Data Set Production

The creation of the VGG16_BN network dataset involves several steps. Firstly, UAV aerial images are utilized for manhole cover acquisition by performing manhole cover positioning. The acquired manhole cover images are then processed to extract the text present on them. Subsequently, the extracted text is inputted into the super-resolution reconstruction network for the purpose of reconstruction. In this study, a total of 3249 manhole cover text images were collected, consisting of 1530 images from rainwater manhole covers and 1719 images from sewage manhole covers. The dataset was partitioned using the K-fold cross-validation method, with a value of K set to 5 for the division.

### 3.3. Evaluation Parameters and Indicators

This study assessed the performance of the manhole cover localization network utilizing metrics such as precision, recall, and mean average precision (mAP). The image super-resolution reconstruction was evaluated using peak signal-to-noise ratio (PSNR) and a mean squared error (MSE) as the benchmark indices. Additionally, the text classification network was gauged through parameters including precision, recall, and F1_Score. Detailed computational methods for these indicators are delineated below.

$$Precision = \frac{TP}{TP + FP} \tag{1}$$

$$Recall = \frac{TP}{(TP + FN)} \tag{2}$$

$$mAP = \frac{\sum_{i=1}^{n-1} AP_i}{k} \tag{3}$$

$$top - 1\ accuracy = \frac{TP + TN}{TP + FP + FN + TN} \tag{4}$$

$$F1\_Score = 2 \times \frac{Precision \times Recall}{Precision + Recall} \tag{5}$$

$$SSIM(x,y) = \frac{(2\mu_x\mu_y + c_1)(2\sigma_{xy} + c_2)}{(\mu_x^2 + \mu_y^2 + c_1)((\sigma_x^2 + \sigma_y^2 + c_2))} \tag{6}$$

$$PSNR = 20 \times log_{10} \frac{MAX_I}{\sqrt{MSE}} \tag{7}$$

In this study, as shown in Figure 14. TP (true positive) quantifies the number of instances where the model accurately identifies actual positive cases as such. Conversely, FP (false positive) counts the instances where the model erroneously labels actual negative cases as positive. FN (false negative) tabulates the number of cases in which actual positive instances are misclassified as negative, while TN (true negative) records the correct classification of actual negative cases. Here, $x$ denotes the original image, and $y$ signifies the reconstructed image. $\mu_x$ and $\mu_y$ are the mean values of the respective images, while $\sigma_x$ and $\sigma_y$ represent their variances. $\sigma_{xy}$ is the covariance between $x$ and $y$. The constants $c_1$ and $c_2$ are utilized to maintain computational stability. Precision is the metric that quantifies the ratio of true positive predictions out of all positive predictions made by the model. Recall assesses the fraction of relevant instances that the model correctly identifies out of the total number of actual positives, reflecting the model's sensitivity. The mean average precision (mAP) is an aggregate measure that computes the average of the individual mean average precisions across multiple categories, with a value range from 0 to 1. A higher mAP value indicates superior model precision across various classes. Top-1 accuracy is the metric that evaluates the model's overall predictive accuracy by measuring the ratio of the number of correctly predicted samples to the total number of samples. F1 score is an evaluation index that considers both precision and recall. The structural similarity index (SSIM) serves as an indicator for assessing the perceptual similarity between two images.

The peak signal-to-noise ratio (PSNR) is employed to compare the maximum potential power of the original signal against the power of distortion noise, thereby quantifying the reconstruction quality of the image.

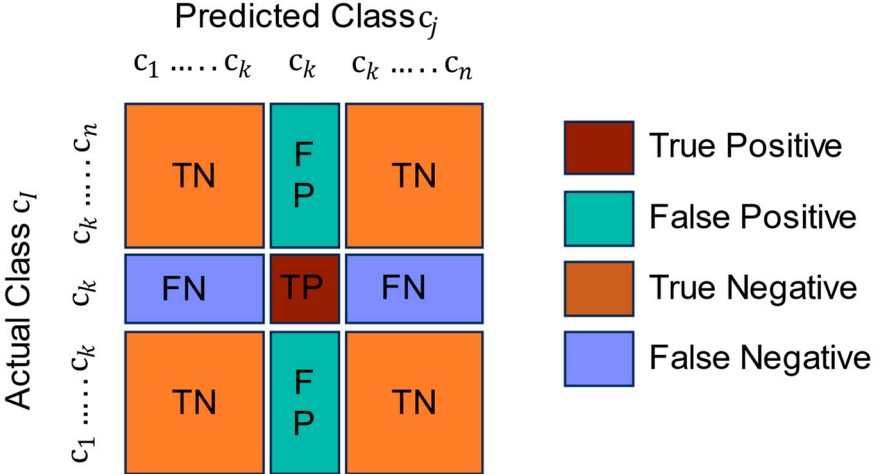

**Figure 14.** Evaluation index diagram.

### 3.4. Setting of Training Parameters

The Pytorch framework was used to train the YOLOv8 network, SRGAN network, and VGG16_BN network. The learning rate of YOLOv8 was $1 \times 10^{-2}$, the optimizer used SDG, the loss function was the cross entropy loss function, the training batch was set to 50, the batch size was set to 16, momentum was 0.937, the reconstruction size of SRGAN network was 4, the learning rate was $2 \times 10^{-4}$, the optimizer was Adam, the training batch epoch was set to 300, the batch size was 8, and the loss function was L1 Loss. The learning rate of VGG16_BN was $1 \times 10^{-2}$, the training batch Epoch was set to 100, the batch size was 32, and the optimizer used SGD.

## 4. Results

### 4.1. Analysis of Manhole Cover Positioning Results

The data set was input into three target detection networks for training, and the cover positioning network obtained the training results after 50 iterative trainings. The results are shown in Table 2.

**Table 2.** Indicators for evaluating manhole cover location networks.

| Evaluating Indicator | YOLOv8 | Faster R-CNN | EfficientNet |
|---|---|---|---|
| Precision | 97.41% | 93.01% | 96.88% |
| Recall | 97.13% | 99.43% | 97.14% |
| mAP | 99.63% | 96.84% | 97.33% |
| F1_Score | 97.27% | 96.11% | 97.01% |

By analyzing the values of the evaluation indicators in the table, it can be clearly observed that YOLOv8 performs better than other networks in the manhole cover positioning task. In particular, mAP was used as the evaluation index of positioning error, and the score of YOLOv8 was as high as 99.63%, indicating that YOLOv8 had excellent positioning accuracy. This result was helpful for the accuracy of subsequent manhole cover cutting. From the precision point of view, YOLOv8 also had a good performance. This meant that YOLOv8 could identify the manhole cover target with high accuracy.

In addition, by analyzing Figure 15c,d, it could be seen that the YOLOv8 network could also locate the incomplete manhole cover well. From the Figure 15e, it could be seen that the YOLOv8 network could also accurately locate the manhole cover after it was

blocked by stains. At the same time, it could be seen from Figure 15b,e that YOLOv8 also had good recognition ability for manhole covers of different shapes and sizes, and can accurately locate.

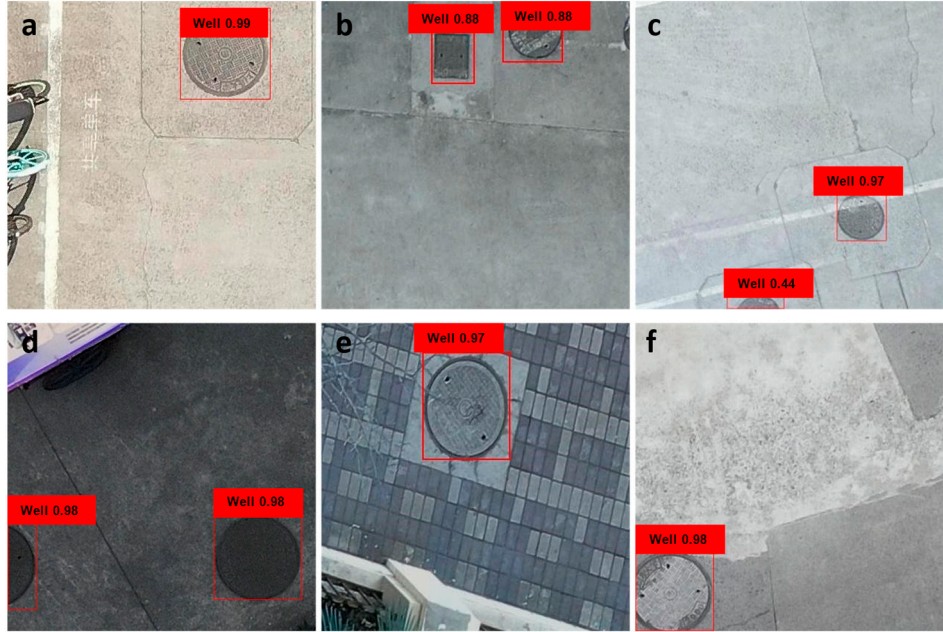

**Figure 15.** Display of the results of manhole cover localization. (**a**) sunny day shot of manhole covers (**b**,**e**) cloudy day shot with different types of manhole covers (**c**,**d**) mutilated manhole covers (**f**) stained cover manhole covers.

### 4.2. Analysis of Text Positioning Results

The text localization dataset was fed into the YOLOv8 network for training, and the training results of the text localization network were obtained through 50 iterations of network training, as shown in Table 3.

**Table 3.** YOLOv8 text positioning results.

| Evaluating Indicator | YOLOv8 |
|---|---|
| Precision | 98.54% |
| Recall | 96.31% |
| mAP | 99.76% |
| F1_Score | 97.41% |

Table 3 showed that YOLOv8 performs well in text localization, with an average precision of 99.76%. This indicated accurate localization of almost all manhole cover texts. At the same time, the network's recall rate reached 96.31%, indicating that most of the manhole cover texts were successfully localized. From Figure 16, it could be seen that the YOLOv8 network could accurately locate the text in the incomplete manhole covers a and f, and it could also be well positioned for the manhole covers covered by stains. In addition, for the square manhole covers a and b, YOLOv8 could also be well positioned. The manhole covers d and c under different weather and light conditions could also be accurately positioned by the network. Therefore, text positioning through the YOLOv8 network can accurately obtain the position of the text.

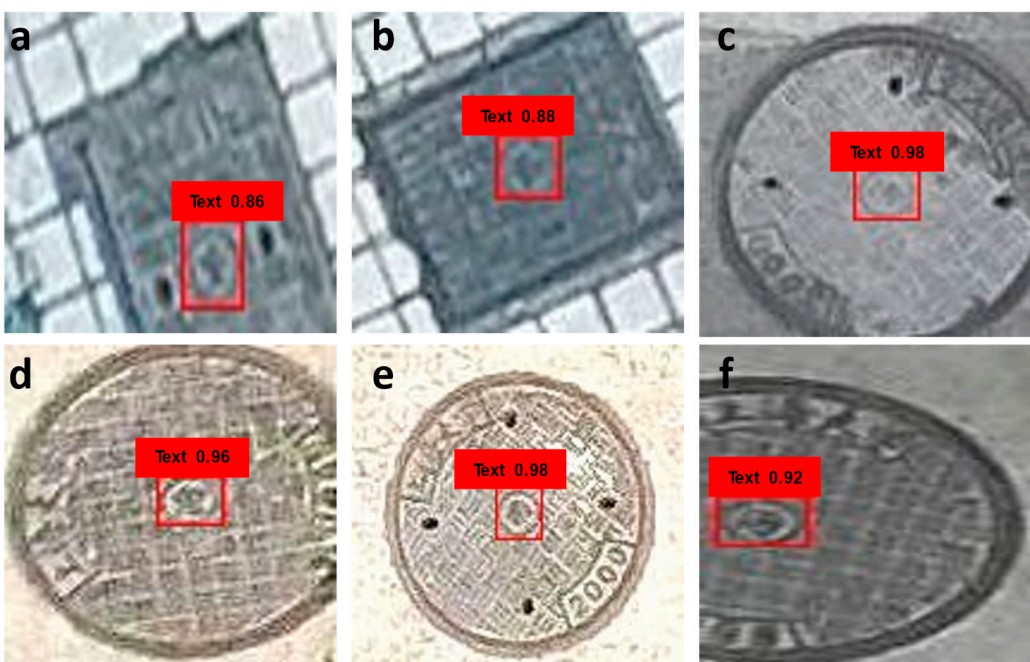

**Figure 16.** Text positioning results display. (**a**,**b**) square manhole covers (**c**) stained covered manhole covers (**d**,**e**) sunny shooting manhole covers (**f**) mutilated manhole covers.

*4.3. Super-Resolution Reconstruction of Morehole Cover Text*

The processed dataset was fed into the SRGAN network for training iterations of 300 epochs to obtain the results of the super-resolution reconstruction network for manhole covers. The specific training parameters are listed in Table 4.

**Table 4.** SRGAN reconstruction effect evaluation index table.

| Evaluating Indicator | SRGAN |
|:---:|:---:|
| PSNR | 29.54 |
| SSIM | 0.83 |

From Table 4, it was known that SRGAN achieved higher PSNR values in the super-resolution reconstruction of the manhole cover training. It showed that the reconstructed image had less visual error with the original high-resolution image. It could also be seen from the Figure 17 that the reconstructed image could well reconstruct the low resolution image to a clear high resolution image. SRGAN also obtained a high SSIM value, which reflected that the reconstructed image preserves the details of the original image well. The images had a high similarity in terms of texture, structure, and brightness. In general, manhole cover reconstruction using SRGAN network is an effective method.

*4.4. Analysis of Manhole Cover Classification Results*

In this paper, Mobilenetv1, Swin_transformer_tiny and VGG16_BN networks were trained, and the results were shown in Table 5. Comparing the three networks, VGG16_BN had the best effect in the classification of manhole covers, the recognition accuracy of manhole covers was 97.62%, and the comprehensive score was the highest in the three networks, so VGG16_BN was selected as the classification network of manhole covers in this paper.

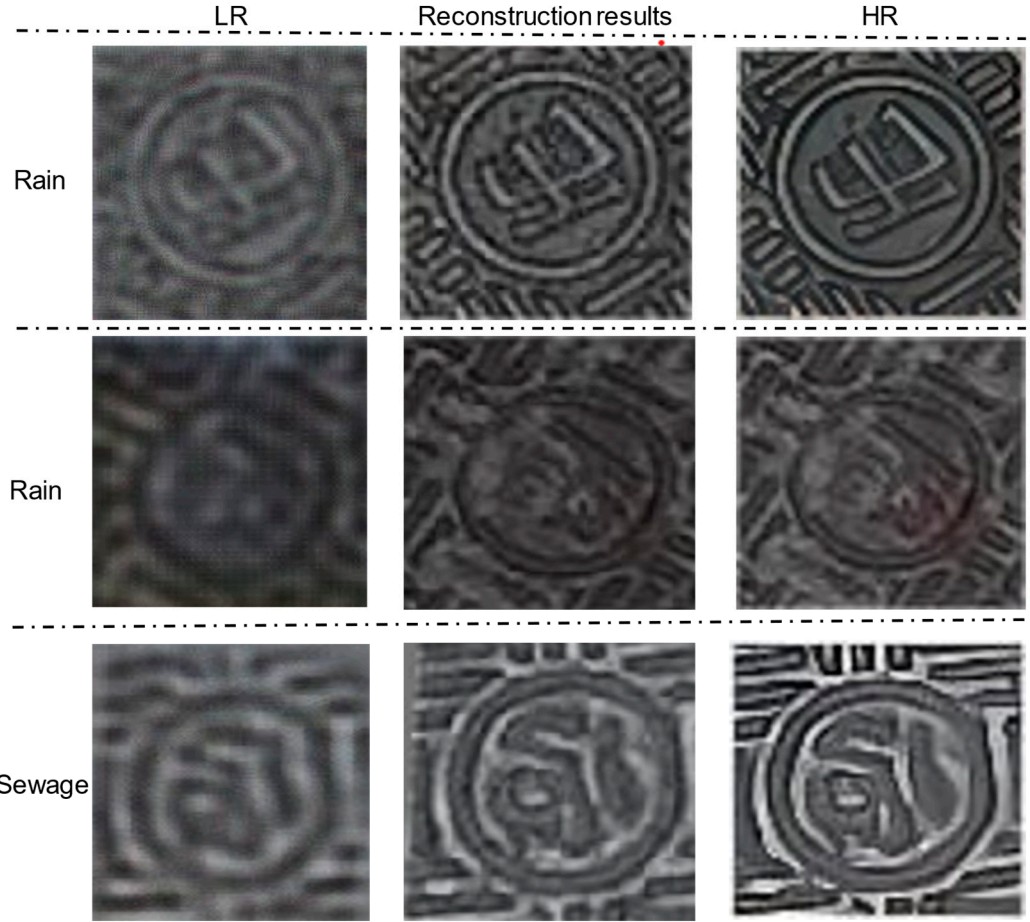

**Figure 17.** Super-resolution reconstruction results.

**Table 5.** Comparative evaluation metrics for well-cover categorization networks.

| Evaluating Indicator | Mobilenetv1 | Swin_transformer_tiny | VGG16_BN |
|---|---|---|---|
| Precision | 92.73% | 90.0% | 97.62% |
| Recall | 92.73% | 94.32% | 98.86% |
| mAP | 93.75% | 92.19% | 98.44% |
| F1_Score | 92.73% | 92.11% | 98.24% |

After determining the image classification network, the VGG16_BN network undergoes K-value cross-validation to verify the consistency and reliability of the classification results. The verification results were presented in Table 6. It could be observed from the table that following five-fold cross-validation, the network demonstrated consistent performance, with an accuracy rate hovering at approximately 93.92%. Based on the K-fold calculation results, it could be concluded that at a 95% confidence level, the confidence interval of the classification accuracy was [0.93, 0.95] with a confidence coefficient of 1.96. This indicated the reliability of the results obtained from the VGG16_BN network.

After determining the classification network, the hyperparameters need to be determined. In order to evaluate the influence of different learning rates on the model, this experiment used $1 \times 10^{-2}$, $1 \times 10^{-3}$, and $2 \times 10^{-5}$ to train the model for 150 epochs, and collected the loss function during the training process. Through Figure 18, it could be seen that the learning rate of $1 \times 10^{-2}$ is the fastest model convergence, and the model tends to be stable at 75 epochs. Therefore, the learning rate is set to $1 \times 10^{-2}$. The training batch is 100 epochs.

**Table 6.** K-fold cross-validation results.

| Evaluating Indicator | Precision | Recall | mAP | F1_Score |
|---|---|---|---|---|
| K1 | 94.96% | 94.42% | 95.18% | 94.69% |
| K2 | 94.35% | 94.54% | 93.97% | 94.44% |
| K3 | 92.02% | 92.02% | 92.26% | 92.02% |
| K4 | 93.15% | 93.22% | 92.52% | 93.18% |
| K5 | 95.12% | 93.71% | 95.48% | 94.41% |
| Mean Value | 93.92% | 94.06% | 94.21% | 94.49% |
| Mean Square Deviation | 0.012 | 0.014 | 0.018 | 0.010 |

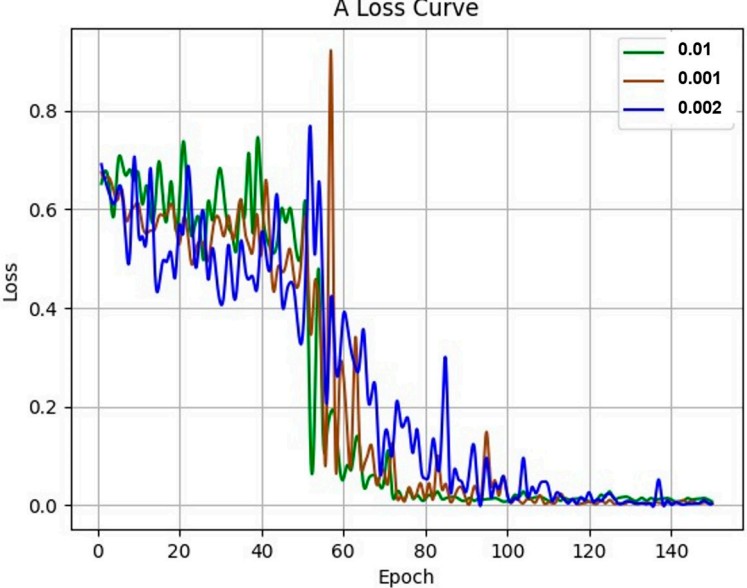

**Figure 18.** Hyperparameter training loss results.

After determining the parameters, the classification performance of the reconstructed manhole cover text images was validated. Under the condition of keeping other parameters consistent, the unreconstructed manhole cover text images and the reconstructed manhole cover text images were separately inputted into the network for training. By comparing the precision, recall, and top-1 accuracy metrics after training, the effectiveness of super-resolution reconstruction was evaluated. The final training evaluation metrics are presented in Table 7.

**Table 7.** Indicators for evaluating manhole cover categorization networks.

| Method | Precision | Recall | Top-1 Accuracy | F1_score |
|---|---|---|---|---|
| Original images | 84.92% | 77.95% | 82.81% | 81.29% |
| Super-resolution reconstruction | 97.62% | 98.86% | 98.44% | 98.24% |

It could be seen from Table 7 that all aspects of the VGG16_BN network trained with the super-resolution reconstructed manhole cover had been greatly improved. Among them, the recall rate of the image without super-resolution reconstruction was only 77.95%. After training with the super-resolution reconstruction data set, the recall rate had increased by 20%. At the same time, it could be seen from Figure 19 that even if it was tested on the original image, the network trained by super-resolution had also achieved a significant accuracy improvement. It could be concluded that the network trained by super-resolution reconstruction has a significant improvement in the accuracy of manhole cover classification.

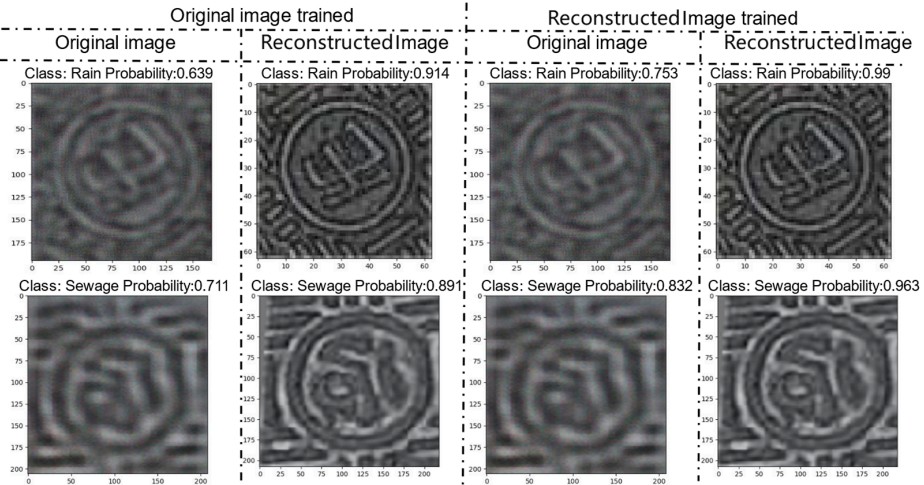

**Figure 19.** The result of super-resolution image text recognition.

In addition, the robustness of the VGG16_BN network was verified by testing under different weather and different light conditions. The results were shown in Figure 20. It could be seen from the diagram that the VGG16_BN network could accurately identify the type of manhole cover, whether it was cloudy or sunny. At the same time, for the manhole cover text c and d under different light intensity, the network could also accurately identify the category of the manhole cover. It could be seen that the VGG16_BN network could accurately identify the category of the manhole cover under different weather and light intensity.

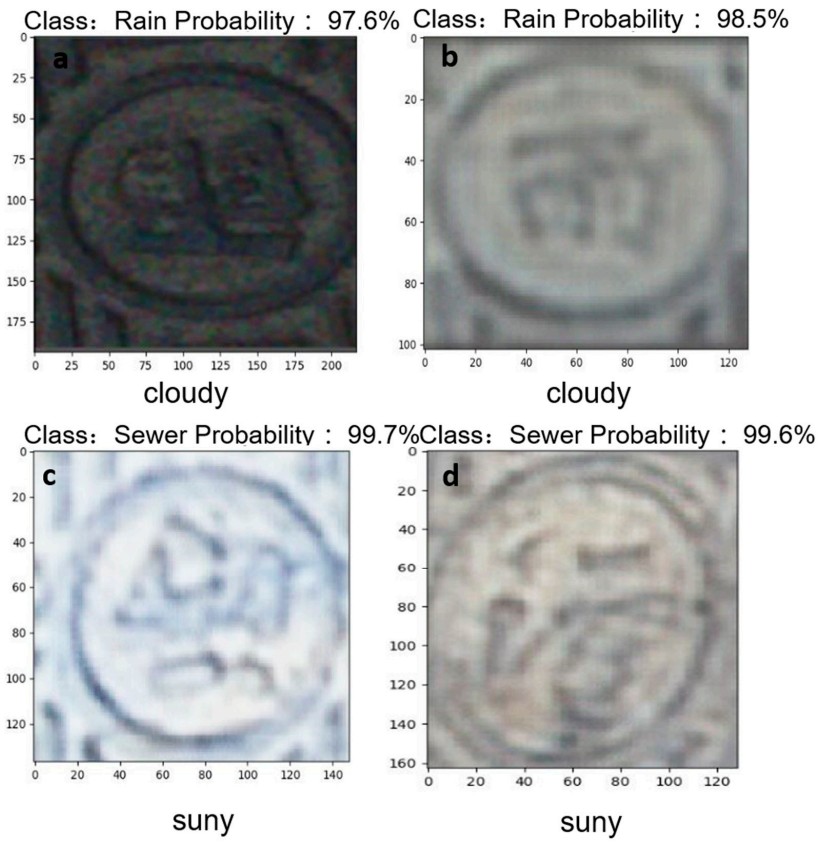

**Figure 20.** Character recognition network training result diagram. (**a**) low-brightness manhole cover (**b**–**d**) high-brightness manhole cover.

### 5. Conclusions

To address the issue of missing information in old urban manhole covers, this paper proposed a manhole cover classification and text recognition method based on an image super-resolution reconstruction network. The method possessed the following functionalities:

1.  The experimental results showed that it was an effective method to locate the manhole cover and text before text recognition.
2.  The experiment proved that the accuracy of manhole cover classification could be improved by using the method of image super-resolution reconstruction to clarify the cap text and reconstruct the aerial image with missing text details due to the long aerial distance.
3.  The method of using VGG16_BN to classify manhole covers is an effective method. It can accurately identify the type of manhole cover, and the recognition accuracy is as high as 97.62%.

Experimental results had demonstrated the feasibility of the algorithm proposed in this study for addressing the issue of missing information in manhole covers in old urban areas. The results indicated that the proposed method can effectively perform manhole cover localization and classification. This method holds significant importance for collecting information on the underground pipe network in urban infrastructure management. By accurately obtaining the category information of manhole covers, city network administrators can better maintain underground facilities. However, this method also has limitations, as manhole covers that were obstructed during aerial imaging or lack textual category information cannot have their category determined. Further research is needed to address and overcome this limitation.

**Author Contributions:** Conceptualization, Y.H. and D.W.; methodology, Y.H.; software, D.W.; validation, Y.H.; formal analysis, Y.H.; investigation, Y.H.; resources, D.W.; data curation, Y.H.; writing—original draft preparation, Y.H.; writing—review and editing, D.W.; visualization, Y.H.; supervision, D.W. All authors have read and agreed to the published version of the manuscript.

**Funding:** This research received no external funding.

**Institutional Review Board Statement:** Not applicable.

**Informed Consent Statement:** Not applicable.

**Data Availability Statement:** The data presented in this study are available at https://drive.google.com/drive/folders/1H6awVOf2P_2ATZunb8d3x_h754wCaeGA?usp=sharing.

**Conflicts of Interest:** The authors declare no conflicts of interest.

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
