# Peer review of "Manhole Cover Classification Based on Super-Resolution Reconstruction of Unmanned Aerial Vehicle Aerial Imagery"

_applsci, doi:10.3390/app14072769_

Round 1

Reviewer 1 Report

Comments and Suggestions for Authors

The proposed method for manhole cover detection based on UAV aerial photography offers an interesting solution by leveraging image super-resolution reconstruction and image positioning and classification techniques. There are however major points for improvement:

Authors must evaluate the performance of the proposed method under different environmental conditions such as varying lighting conditions, weather conditions (e.g., sunny, cloudy, rainy), and times of day.

Authors must assess the robustness of the method against noise and interference commonly encountered in urban environments, including noise from traffic, pedestrians, and other urban activities.

Authors must test the method's robustness against different types of manhole covers in terms of size, shape, color, and text/font variations.

Authors must perform cross-validation to assess the consistency and reliability of the classification accuracy results across different subsets of the dataset.

Authors must validate the proposed method's performance using appropriate statistical tests (e.g., t-tests, ANOVA) to determine if the classification improvement is statistically significant.

Authors must calculate confidence intervals for the reported classification accuracy to provide a measure of uncertainty and reliability of the results.

Authors must evaluate the impact of removing or disabling specific components or stages of the proposed method (e.g., object detection, super-resolution processing, classification) on the overall performance to assess the contribution of each component to the final results.

Authors must analyze the sensitivity of the method's performance to key parameters such as detection threshold, super-resolution factor, and classifier settings by systematically varying these parameters and observing their effects on classification accuracy.

Authors must compare the performance of different variants of the proposed method by modifying or replacing individual components with alternative methods (e.g., using different object detection models, super-resolution algorithms, or classification networks) to identify the most crucial components for achieving high classification accuracy.

Author Response

See annex for details

Reviewer 2 Report

Comments and Suggestions for Authors

Dear all,

The idea of the paper is interesting, but I cannot find the purpose of the experiment. It is well known that all the mayors have detailed plans with all components of the underground infrastructure.

There are some issues related to the text of the paper as follows:

- at row 54 - there is a difficult text to understand...,

- figures are not well labeled and cited in text...,

- tables are not well labeled and cited in text...,

- tables 2 and 3 are identical (is it correct?),

- rows 388 and 389 present other values than those from Table 3 (as it is mentioned)...

- row 395 has a typo-error.

I recommend a major revision of the purpose of the article and also of the article itself.

Comments on the Quality of English Language

In general, the paper has some hard to understand phrases (maybe, due to the poor English or due to the lack of explanations).

Author Response

See annex for details

Reviewer 3 Report

Comments and Suggestions for Authors

Manuscript ID: applsci-2870218

Title: Manhole Cover Classification Based on Super-Resolution Reconstruction

of Unmanned Aerial Vehicle Aerial Imagery

Authors: Dejiang Wang *, Yuping Huang

Congratulations on your work titled: Manhole Cover Classification Based on Super-Resolution Reconstruction of Unmanned Aerial Vehicle Aerial Imagery. However, I have the following questions and suggestions

1.- How many images are considered sufficient to achieve good training of the network to generate a good quality image?

2.- For this study, is the F1-score advisable to use?

3.- Some works have used similar neural networks, the percentage results for precision, recall and mAP are closer to 100%. Is it possible to adjust the hyperparameters, or the neural training to improve the percentage results?

4.- How efficient does this methodology become if in the images obtained by the unmanned aerial vehicle you obtain images with dirt or garbage on the manhole cover?

5.- You say that the method used in this work is feasible and robust, however, the question is, what is the robustness test that was used to mention it?

Author Response

See annex for details

Round 2

Reviewer 1 Report

Comments and Suggestions for Authors

Paper was improved and can now be accepted after small modifications regarding the robustness, which is still not very clear, but at least it is somewhat covered now (minor review).

Author Response

See annex for details

Reviewer 3 Report

Comments and Suggestions for Authors

Congratulations to the authors

Kind regards

Author Response

Thank you very much for your review and I wish you all the best!